# Role of oxidative metabolism in osseointegration during spinal fusion

**Laura C. Shum**[ɵ], **Alex M. Hollenberg**[ɵ], **Avionna L. Baldwin, Brianna H. Kalicharan, Noorullah Maqsoodi, Paul T. Rubery, Addisu Mesfin, Roman A. Eliseev**[ID] *

Center for Musculoskeletal Research, University of Rochester Medical Center, Rochester, NY, United States of America

ɵ These authors contributed equally to this work.
* roman_eliseev@urmc.rochester.edu

**Data Availability Statement:** All relevant data are within the manuscript.

## Abstract

Spinal fusion is a commonly performed orthopedic surgery. Autologous bone graft obtained from the iliac crest is frequently employed to perform spinal fusion. Osteogenic bone marrow stromal (a.k.a. mesenchymal stem) cells (BMSCs) are believed to be responsible for new bone formation and development of the bridging bone during spinal fusion, as these cells are located in both the graft and at the site of fusion. Our previous work revealed the importance of mitochondrial oxidative metabolism in osteogenic differentiation of BMSCs. Our objective here was to determine the impact of BMSC oxidative metabolism on osseointegration of the graft during spinal fusion. The first part of the study was focused on correlating oxidative metabolism in bone graft BMSCs to radiographic outcomes of spinal fusion in human patients. The second part of the study was focused on mechanistically proving the role of BMSC oxidative metabolism in osseointegration during spinal fusion using a genetic mouse model. Patients' iliac crest-derived graft BMSCs were identified by surface markers. Mitochondrial oxidative function was detected in BMSCs with the potentiometric probe, CMXRos. Spinal fusion radiographic outcomes, determined by the Lenke grade, were correlated to CMXRos signal in BMSCs. A genetic model of high oxidative metabolism, cyclophilin D knockout (CypD KO), was used to perform spinal fusion in mice. Graft osseointegration in mice was assessed with micro-computed tomography. Our study revealed that higher CMXRos signal in patients' BMSCs correlated with a higher Lenke grade. Mice with higher oxidative metabolism (CypD KO) had greater mineralization of the spinal fusion bridge, as compared to the control mice. We therefore conclude that higher oxidative metabolism in BMSCs correlates with better spinal fusion outcomes in both human patients and in a mouse model. Altogether, our study suggests that promoting oxidative metabolism in osteogenic cells could improve spinal fusion outcomes for patients.

## Introduction

Spinal fusion has become one of the most commonly performed orthopedic procedures to treat spinal instability (e.g. due to trauma or tumor) and degenerative conditions of the spine

**Funding:** Funding for this project was provided by a grant from the Orthopaedic Research and Education Foundation with additional funding provided by Musculoskeletal Transplant Foundation, NIH/NIAMS R01 AR072601 grant to RAE, grant from J. Robert Gladden Society to AM, and NIH/ NCATS TL1-TR000096 grant to LCS. The funders had no role in study design, data collection and analysis, decision to publish, or preparation of the manuscript.

**Competing interests:** The authors have declared that no competing interests exist.

**Abbreviations:** BMSCs, Bone marrow stromal (a.k.a. mesenchymal stem) cells; OxPhos, oxidative phosphorylation; CypD, cyclophilin D; BMP, bone morphogenic protein; PBS, phosphate buffered saline; CFU, colony forming unit; ALP, Alkaline phosphatase; NBF, neutral buffered formalin; micro-CT, micro-computed tomography; BMD, micro-computed tomography; OCR, oxygen consumption rate; MPTP, mitochondrial permeability transition pore.

(i.e. scoliosis, spondylolisthesis, or disc herniation) [1,2]. In a recent analysis of the United States National Inpatient Sample (NIS), it was reported that the volume of elective lumbar fusion procedures for degenerative diagnoses increased from 122,679 cases in 2004 to 199,140 in 2015, representing a 62.3% increase [3]. Current practice utilizes spinal instrumentation, such as rods, screws, and plates, to temporarily stabilize the joint, combined with a bone graft to promote intervertebral bony fusion, thus creating permanent biologic stability. Autologous bone graft (autograft) harvested from the patient's iliac crest is currently the gold standard for spinal fusion [4,5]. Autograft promotes osseous formation by contributing osteoconductive materials (e.g. bone minerals and collagens that provide structural support), osteoinductive factors (e.g. bone promoting signals such as cytokines and growth factors), and osteogenic cells (e.g. marrow-derived osteoblastic and preosteoblastic stem cells) to the fusion bed [6,7]. Unfortunately, even with an autograft, the outcome of spinal fusion is not always optimal in older patients, post-menopausal women, and sometimes patients with no obvious confounding conditions [8–10]. Pseudoarthrosis, or failed intervertebral bony fusion, is a well-reported complication of spine surgery [11–16]. The etiology is not often clear, however important risk factors include smoking, diabetes, osteoporosis, and long-term steroid use [17–20]. Multilevel fusions are also known to be associated with a higher incidence of pseudoarthrosis [21]. Although not always symptomatic, patients with pseudoarthrosis can present with spinal instability, recurrent pain, and/or neurological symptoms at long-term follow-up, often requiring revision surgery [22–24]. Therefore, in order to reduce the burden to both patients and the health care system, there is a clinical need to enhance the efficacy of spinal fusion procedures.

Several biological enhancement strategies to promote successful spinal fusion have been proposed in animal and human clinical studies. These include use of bisphosphonates, teriparatide (recombinant deoxyribonucleic acid form of parathyroid hormone [PTH1-34]), prostaglandin agonists, bone morphogenic proteins (BMPs), and stem cells [25]. Among these, bone marrow stromal (a.k.a. mesenchymal stem) cells (BMSCs) have gained considerable attention. Comprising less than 0.01% of the bone marrow cell population, BMSCs are present in autograft and are believed to be the primary driver of new bone formation during spinal fusion [7]. They are multipotent stem cells that have self-renewal capability and can differentiate into osteoblasts to promote bony fusion [26,27]. BMSCs are characterized by their ability to adhere to plastic in culture media and display specific cell surface markers (e.g. cd31-, cd45-, cd105+) [7,28]. BMSCs rely on local paracrine signaling, including BMP/SMAD, Wnt/β-catenin, and Notch, to regulate osteoblast differentiation [29]. Besides being osteoprogenitors, BMSCs also stimulate angiogenesis and innervation [30], important components to the bone healing and regeneration process. As such, it is conceivable that the quality of patient-derived BMSCs is an important factor to both spinal fusion outcome and osseointegration.

There is a growing body of literature indicating that bone deterioration and delayed bone healing are due to a decreased osteogenic potential of BMSCs [31–33]. We and others have reported that BMSCs rely on glycolysis in an undifferentiated state and activate mitochondrial oxidative phosphorylation (OxPhos) during osteogenic differentiation [34–36]. In aging, BMSC mitochondrial function is impaired [37], and our previous work has shown that decreased bone formation correlates with decreased oxidative metabolism in aged bone [38]. Most recently, we reported that improving mitochondrial function via inhibition of the mitochondrial permeability transition pore can improve bone healing [39]. Together, these data posit oxidative metabolism in BMSCs as a potential factor affecting osseointegration during spinal fusion. The goal of this study was to assess the correlation between oxidative metabolism of BMSCs and spinal fusion outcome. To our knowledge, we are the first to show that oxidative metabolism in BMSCs correlate with and potentially play a causal role in spinal fusion outcomes. Prospectively evaluating the quality and anabolic potential of autograft containing

BMSCs prior to surgery could prove to be an effective strategy to accurately predict fusion success and promote successful outcomes.

## Methods

### Materials

All reagents were from Sigma (St. Louis, MO) unless otherwise noted.

### Patient population

The human subject study was approved by the University of Rochester Internal Review Board (approval number RSRB00052621). Male and female patients with no history of diabetes (except for one that was diet controlled), cancer, or other chronic illnesses who were scheduled to undergo a spinal fusion procedure were recruited during pre-operative appointments (Table 1). The age range was from 12 to 73 years old. Appropriate written consent was obtained by the treating surgeon. When minors were enrolled, a written consent was obtained from their parents or guardians. Some patients, including several over the age of 80, received off label BMP-2 peri-operatively to stimulate spinal fusion. Such patients were excluded from this analysis, because of potential masking effects produced by BMP-2 treatment. All samples were coded, and data were received and analyzed in a blinded manner and then matched to individual patients.

### Patient bone marrow samples

Bone marrow was obtained via aspiration from the iliac crest after superficial dissection, but before any graft was harvested. Under direct visualization, marrow was aspirated through a 16-gauge needle using a 10 cc syringe. The surgeon changed the site of aspiration so that no more than 3 cc was aspirated from any one site in the iliac bone. Bone marrow samples from the iliac crest were immediately placed in sodium citrate-treated collection tubes. Aliquots of whole bone marrow were reserved for CMXRos staining, and the remainder was spun at 1,500 rpm for 10 min to separate the serum. Serum was removed and the remaining pellet was re-suspended in Red Blood Cell Lysis Buffer (RBC Lysis Solution, QIAGEN, Hilden, Germany) and incubated at room temperature for 10 min. Samples were then strained through a nylon 70-micron cell strainer and spun for 5 min at 1,500 rpm. The supernatant was aspirated, and the pellet was re-suspended in low glucose DMEM (LG-DMEM, Gibco, Gaithersburg, MD). Cells were then counted using a hemocytometer and plated at a required density. After 24 hrs, cells were rinsed 3 times with phosphate buffered saline (PBS) to remove unattached cells and

Table 1. Patient information.

| Patient information | | | |
|---|---|---|---|
| Age, Average (Min-Max) | 51.96 (12–73) | | |
| Gender (M:F) | 12:10 | | |
| Ethnicity | White: 78% | Black: 11% | Other: 11% |
| No. of patients with Diabetes | 1 (Diet-controlled) | | |
| Avg. No. fused segments | 1.77 | | |
| Diagnoses | Spondylolisthesis, Spinal Stenosis, Scoliosis, Radiculopathy, Foraminal Stenosis | | |
| Medications | Carbamazepine, Cetirizine, Cimetidine, Clonazepam, Fenofibrate, Furosemide, Hydrochlorothiazide, Hydralazine, Ibuprofen, Levothyroxine, Methocarbamol, Metoprolol, Nicotine | | |

fresh media was added. Half of the media in each dish was replaced weekly until cells were ready to be used in subsequent experiments.

## Colony formation

Colony forming unit (CFU) assay was performed, as this is a widely accepted method to assess the number of cells in a population having stem- and progenitor-like colony-forming properties [40]. For the CFU assay, cells were plated at 25,000 cells per $cm^2$, grown for 14 days, and then stained with crystal violet for CFU-fibroblastic (CFU-F) [39] or ALP-specific stain for CFU-osteoblastic (CFU-O). *Alkaline phosphatase (ALP) staining*: Cells were rinsed twice with PBS, then fixed in 10% neutral buffered formalin (NBF) for 2–3 min. NBF was rinsed with PBS and cells were then stained with NBT/BCIP 1 Step Solution (Thermo Fisher Scientific, Waltham, MA) for 20 min, then rinsed 3 times with PBS, air-dried, and photographed. *Crystal violet staining*: Cells were rinsed twice with PBS and then stained with crystal violet (0.5 g stain in 100 mL 100% methanol) for 20 min. Methanol acts as a fixative, so there is no need for NBF. Cells were then rinsed thoroughly with water, air-dried, and photographed.

## Flow cytometry

Live cells are best detected using surface marker immunolabeling and flow cytometry [41,42]. Surface markers were detected with fluorescently-labeled anti-human or anti-mouse antibodies against CD105, CD31, and CD45 (BD Bioscience, Franklin Lakes, NJ) using flow cytometry and 12-color FACS Canto II machine (BD Bioscience, Franklin Lakes, NJ), as previously described [39].

## CMXRos staining

Mitochondrial OxPhos function can conveniently be assessed via measurement of the mitochondrial inner membrane potential. A variety of potential-sensitive fluorescent probes are available. Of these, CMXRos is the most resistant to treatments and modifications, such as fixation and washing [43]. Therefore, cells were stained with 100 nM CMXRos (± 0.5 μM Antimycin A as a negative control) for 15 min at 37°C, then spun down and re-suspended in cell freezing media (LG-DMEM, 20% FBS, 10% DMSO) and kept at -80°C until the day of the assay. As a negative control, a subset of each sample was pre-incubated with mitochondrial respiratory complex III inhibitor Antimycin A at 0.5 μM before the addition of CMXRos. Antimycin A blocks the respiratory chain and completely depolarizes the mitochondria [44]. On the day of the assay, cells were thawed and labeled with antibodies against CD105, CD45, and CD31 to use for flow cytometry. CMXRos signal in CD105+ CD45- CD31- population was measured.

## Lenke grading

The Lenke classification is a widely used and accepted radiologic grading scale to evaluate posterolateral fusion success [45–48]. At 12 months following spinal fusion surgery, the surgeon (AM), blinded to the results of the BMSC analyses, assessed spinal graft osseointegration by radiographic assessment. A Lenke grade of A–D was given for each patient (Table 2) [46].

**Table 2. Lenke scoring criteria.**

| Lenke Grade |
| --- |
| A–Fused with remodeling and trabeculae |
| B–Graft intact, not fully remodeled and incorporated throughout; no lucencies |
| C–Graft intact but definite lucency at top or bottom of graft |
| D–Definitely not fused with resorption of bone graft and with collapse |

## Animals

Animal husbandry and experiments were performed in accordance with the Division of Laboratory Animal Medicine, University of Rochester, state and federal law, and National Institutes of Health policies. Approval was obtained from the University of Rochester Institutional Animal Care and Use Committee prior to performing the study. Cyclophilin D knockout (CypD$^{-/-}$) mice were a kind gift from Dr. George Porter (University of Rochester). CypD$^{-/-}$ mice were backcrossed to C57BL/6J for five generations and bred in house. Heterozygous (CypD$^{+/-}$) breeders were used to obtain CypD$^{-/-}$ and CypD$^{+/+}$ littermates [39].

## Posterolateral intertransverse lumbar fusion mouse model

A mouse model of spinal fusion was developed previously and reported to produce consistent bony bridges [49]. We therefore chose this procedure with some modifications. The posterolateral lumbar spine fusion surgery was performed from L4 to L6 levels. A 2 cm dorsal midline skin incision was made centered over the L4-L6 spinous processes. A second 2 cm incision was made midline through the dorsolumbar fascia along the spinous processes, and the paravertebral muscles overlying the transverse processes of L4-L6 were separated from the spinous processes. This was performed bilaterally, creating a pocket between muscle and bone on both sides of the spine. The transverse processes were then exposed. A scalpel blade was used to decorticate the visible transverse processes until punctuate bleeding was observed. The bone graft (3 mm x 1 mm piece of iliac crest bone with bone marrow from littermate donor mouse) was then inserted into the pocket under the dorsal muscle next to the spine in contact with the transverse processes of L4-L6 on each side. The dorsolumbar fascia and skin were then sutured.

## Syngeneic mouse bone graft harvest

Iliac crest was harvested from a littermate donor mouse of the same genotype to prevent immune rejection. A dorsal midline skin incision was made to expose the iliac crest, which was then obtained from the donor mouse. Attached tissue was removed from the iliac crest, and one side of the iliac crest bone was cut off/chipped away to expose the bone marrow within the bone. To mimic human autograft procedure, the iliac crest from a control CypD$^{+/+}$ mouse was placed along the transverse processes of a control CypD$^{+/+}$ littermate. The iliac crest from a CypD$^{-/-}$ mouse was placed along the transverse processes of a CypD$^{-/-}$ littermate.

## Micro-computed tomography

At 12 weeks post-surgery, mice were sacrificed. Fusion of the L4-L6 vertebrae was assessed by micro-computed tomography (micro-CT), the most informative method to assess newly formed bone microarchitecture and quality [50,51]. Point of contact between the bone graft and the spine were counted using the 3-dimensional micro-CT images. Total new bone volume was calculated and divided by points of contact. Bone mineralization density (BMD) of the newly formed bone was also calculated.

## Seahorse metabolic analysis of BMSCs in vitro

Live cell measurement of changes in media oxygen and acidification using the Seahorse extracellular flux analyzer is currently the most reliable method of cell bioenergetic profiling [35,52–55]. Following the method of Shares et al [39], primary murine BMSCs were isolated from CypD$^{-/-}$ and control CypD$^{+/+}$ littermates by flushing the marrow of tibiae and femurs. Cells were then washed, pelleted, and resuspended in low glucose DMEM (LG-DMEM)

containing glucose at 5 mM, glutamine at 2 mM, 10% fetal bovine serum (FBS), and 1% peni-cillin-streptomycin, and plated at $20 \times 10^6$ BMSCs per 10 cm$^2$ dish. Media was changed on each of the 3 days following BMSC isolation to rid the culture of non-adherent hematopoietic cells. Media was then changed weekly until reaching 80% confluency, which is when the cells were plated for the appropriate experiments. Oxygen consumption rate (OCR) and extracellular acidification rate (ECAR) were measured using Seahorse XF96 apparatus (Seahorse Bioscience). Cells were plated on Seahorse 96-well plates 48 hrs before experiment at a density of $2 \times 10^4$ cells/well. Immediately before the experiment, media was replaced with the unbuffered DMEM-based XF media containing 5 mM glucose, 1 mM glutamine, 1% serum, and no pyruvate (pH 7.4). The Seahorse XF typical analysis involves injections of several metabolic inhibitors and modifiers allowing subsequent calculation of various bioenergetic indices [39]. The final addition of antimycin A, a mitochondrial respiratory complex III inhibitor, serves as a negative control for the mitochondrial OCR because it terminates mitochondrial respiration.

### Statistical analysis

Data were analyzed using Prism 7 (GraphPad Software) and Spearman correlation. Where needed, mean values and standard errors or deviations were calculated; and statistical significance ($p < 0.05$) was established using Student's $t$-test. Power analysis was performed by pooling variances of all groups, then calculating required sample size using G*Power [56], with an alpha of 0.05, and power set to 0.80. The formula used for power analysis was as follows: $n = \frac{2(Z_a + Z_{1-\beta})^2 \sigma^2}{\Delta^2}$, where n = sample size, $\alpha$ = type I error, $\beta$ = type II error, $\Delta$ = effect size, $\sigma$ = standard deviation, and Z is a constant. According to this analysis, we needed n = 10 for both male and female patients.

## Results

We first evaluated whether spinal fusion outcome correlated with patient age. At 12 months after spinal fusion surgery, the surgeon performed blinded analyses of spinal fusion success and graft osseointegration by radiographic assessment (Fig 1A). The radiograph was given a grade according to the Lenke scale, from A (well fused with remodeling and trabeculae present) to D (no fusion, resorption of bone graft) as summarized in Table 2 [46]. We then correlated the Lenke grade to patient age using the Spearman test. Male (blue symbols) and female (pink symbols) data were analyzed separately in this and all other figures. Our analysis showed that there was no correlation between the age of the patient and the Lenke grade since $r$ value did not reach the required 0.7 or -0.7 and $p$ value was significantly higher than 0.05 (Fig 1B).

The success of a spinal fusion procedure is believed to be dependent on the ability of both resident and grafted BMSCs to form new bridging bone. It is therefore important to prospectively evaluate the number and properties of BMSCs. CFU assay is a convenient method to assess the number of cells with clonogenic potential [57–59]. To determine colony forming ability of patient BMSCs, total number of freshly isolated bone marrow cells was counted, and cells were plated at a density of 25,000 cells/cm$^2$ (Fig 2A) followed by removal of unattached (mostly hematopoietic) cells after 24 hrs. This procedure is known to be selective for colony-forming BMSCs. After two weeks, plates were stained with either Crystal Violet for CFU-fibroblastic (CFU-F) or ALP-specific stain for CFU-osteoblastic (CFU-O), and colonies consisting of more than 50 cells were counted. CFU-F assay shows the number of fibroblastic colonies formed, while CFU-O assay demonstrates how many colonies have osteogenic potential. Our data showed that neither CFU-F nor CFU-O numbers correlated with radiographic outcome (Lenke grade) as illustrated in Fig 2B and 2C. Thus, according to our data, colony formation in vitro is not a good predictor of spinal fusion success.

A)

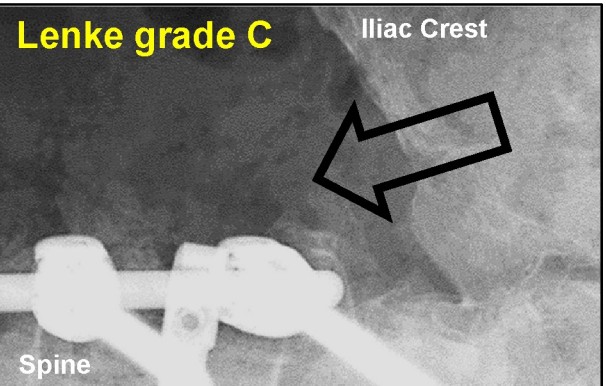

B)

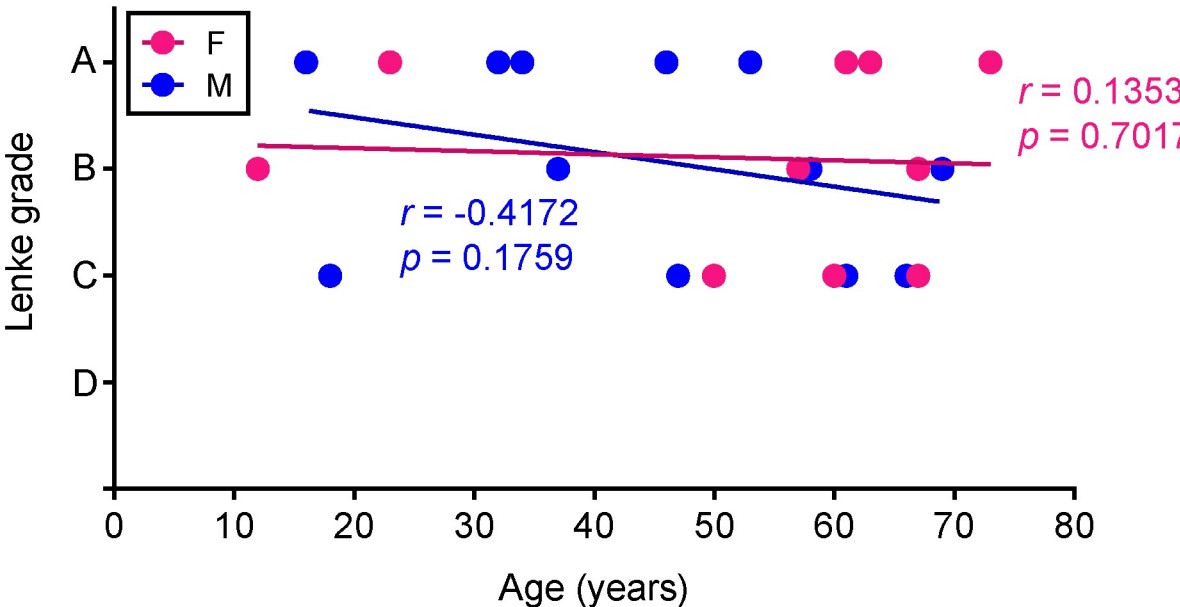

**Fig 1. Radiographic Lenke grade does not correlate with patient age.** A) At 12 months post-surgery, patients' radiograms were blindly assessed for osseointegration using Lenke score from A (most fused) to D (no fusion). Representative X-ray images are shown. Arrows indicate the site of fusion; B) Lenke scores were correlated to age of the patient grouped by sex using the Spearman test. There was no correlation between Lenke grade and age.

As our goal was to evaluate the role of BMSC oxidative metabolism in the success of spinal fusion, we measured mitochondrial function of patient BMSCs via CMXRos staining and flow cytometry. CMXRos is a membrane potential-sensitive fluorescent probe that accumulates primarily in mitochondria due to their high membrane potential and stays unchanged even upon fixation or freezing [60]. CMXRos is therefore not appropriate for dynamic measurements, but is very suitable for our purposes. Mitochondrial membrane potential is critical for mitochondrial ATP production linked to oxidation and is thus a direct measure of mitochondrial OxPhos. As will be shown later in Fig 4, we verified that CMXRos signal correlates with a

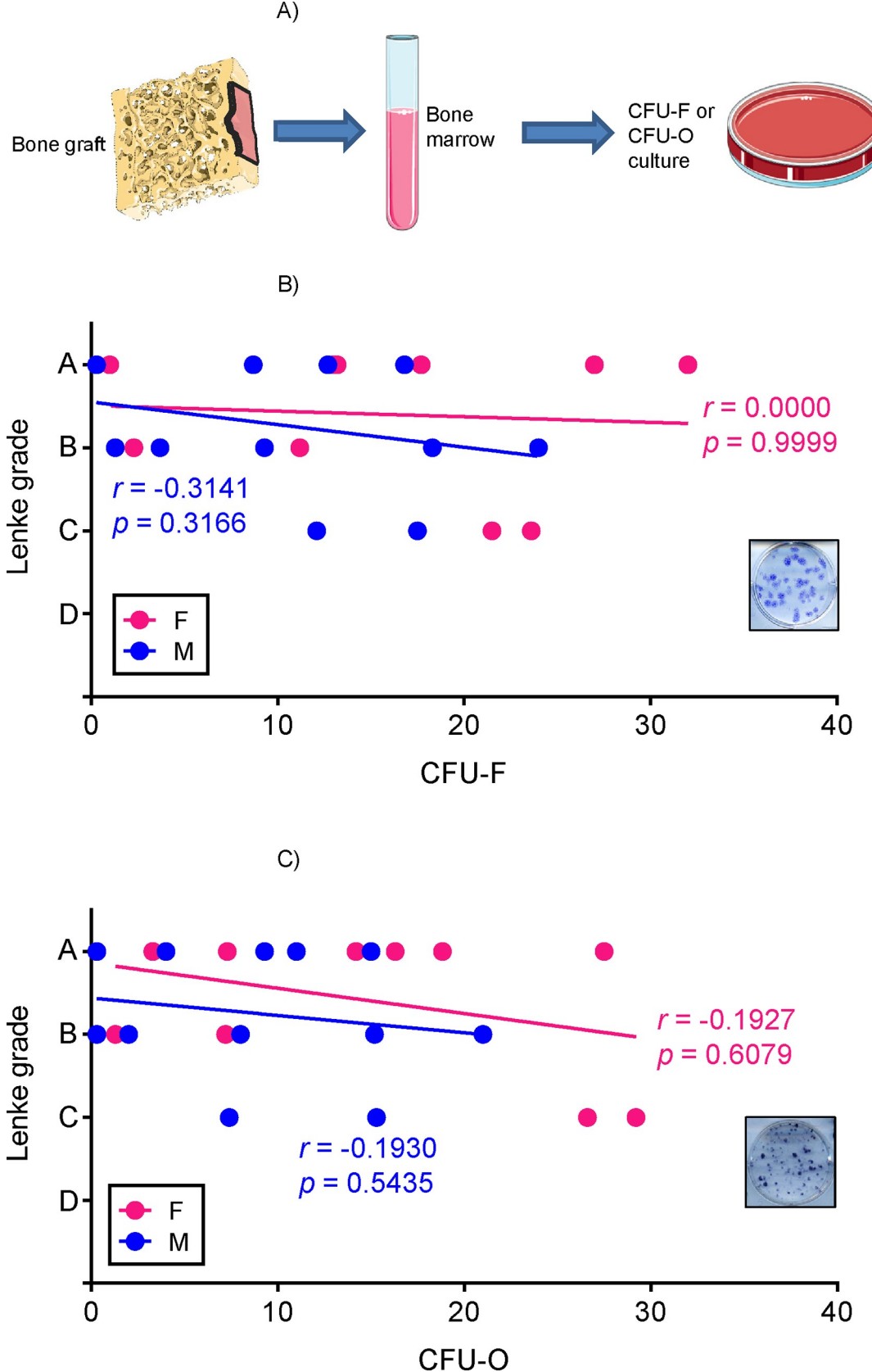

**Fig 2. Radiographic Lenke grade does not correlate with colony forming ability of BMSCs *in vitro*.** A) Bone marrow cells were plated at a density of 25,000 cells/cm$^2$ followed by washout of unattached cells and incubation for 14 days. At D14, colonies were stained with ALP-specific stain (CFU-O) or Crystal Violet (CFU-F) and then correlated to age of the patient grouped by sex. Representative cell culture wells are shown in inserts of panel (B) and (C). There was no correlation by Spearman test between spinal fusion outcomes and either CFU-F number (B) or CFU-O number (C).

more direct measure of OxPhos, cellular Oxygen Consumption Rate (OCR) assayed with the Seahorse XF technology. After staining with CMXRos, whole bone marrow samples were also stained with antibodies against CD31, CD45, and CD105 and analyzed using flow cytometry (Fig 3A). Cells that were negative for CD31 endothelial and CD45 hematopoietic marker and positive for CD105 mesenchymal cell marker were assessed for CMXRos fluorescence levels (Fig 3B). Cells treated with antimycin A, a respiratory complex III inhibitor that terminates oxidative metabolism, served as a negative control. CMXRos signal was plotted against the Lenke grade and Spearman test was run to assess correlations. We observed a notable result when mitochondrial function was compared with spinal fusion outcome. All patients with high CMXRos signal (~400 RFU or higher) and thus high mitochondrial OxPhos function had a favorable spinal fusion outcome, while patients with lower CMXRos signal and thus lower mitochondrial OxPhos function had a less favorable outcome (Fig 3C). Together, these data highlight the importance of mitochondrial oxidative metabolism for BMSC function and implicate the mitochondria as a potential therapeutic target to improve outcomes of orthopedic surgeries (e.g. spinal fusion), which depend on BMSC osteogenic function.

To verify this finding mechanistically, we employed an *in vivo* genetic mouse model. Cyclophilin D (CypD) is the only genetically proven positive regulator of the mitochondrial permeability transition pore (MPTP) in the inner membrane of mitochondria. When this pore is open, mitochondrial OxPhos function declines, however when this pore is kept closed by genetic ablation of CypD, mitochondrial OxPhos function is improved [61,62]. Our previous work on the global CypD knockout model (CypD$^{-/-}$) highlighted the importance of mitochondrial function in bone, and linked the decline in aging bone to a decline in mitochondrial function due to MPTP opening [38]. CypD$^{-/-}$ mice were protected from both a decline in mitochondrial function with age and a subsequent decline in bone mass. In the current study, we used the same CypD$^{-/-}$ mouse model to evaluate whether improved mitochondrial function would improve spinal fusion outcomes. BMSCs from CypD$^{-/-}$ mice and control littermates (CypD$^{+/+}$) were isolated and evaluated for oxidative metabolism using the same approach that was used for human samples, CMXRos staining. We observed that CMXRos signal was significantly higher in CypD$^{-/-}$ BMSCs (Fig 4A). We then verified these data using the Seahorse analysis, which showed that CypD$^{-/-}$ BMSCs had significantly higher OCR (Fig 4B). This indicates more efficient oxidative metabolism in CypD$^{-/-}$ BMSCs when compared to control CypD$^{+/+}$ BMSCs.

To see if the observed difference in oxidative metabolism translates into better outcome of spinal fusion, we performed a spinal fusion surgery in CypD$^{-/-}$ and control CypD$^{+/+}$ mice. Briefly, iliac crest grafts with exposed bone marrow from the donor mice were placed on the left and right transverse spinal processes at the L4-L6 level of littermate recipient mice (Fig 4C). The transverse processes of the recipient mice were abraded beforehand to expose bone marrow. At 12 weeks post-surgery, the mice were evaluated for graft osseointegration and new bone growth at the bridges between the graft and the transverse processes by micro-CT (Fig 4C). Both groups of mice generated a similar amount of bridging bone volume (Fig 4D), however, the bridging bone generated in the CypD$^{-/-}$ mice had a greater level of BMD than the control CypD$^{+/+}$ counterparts (Fig 4E). Collectively, these results show that BMSCs with higher oxidative metabolism created more mineralized bone following spinal fusion surgery.

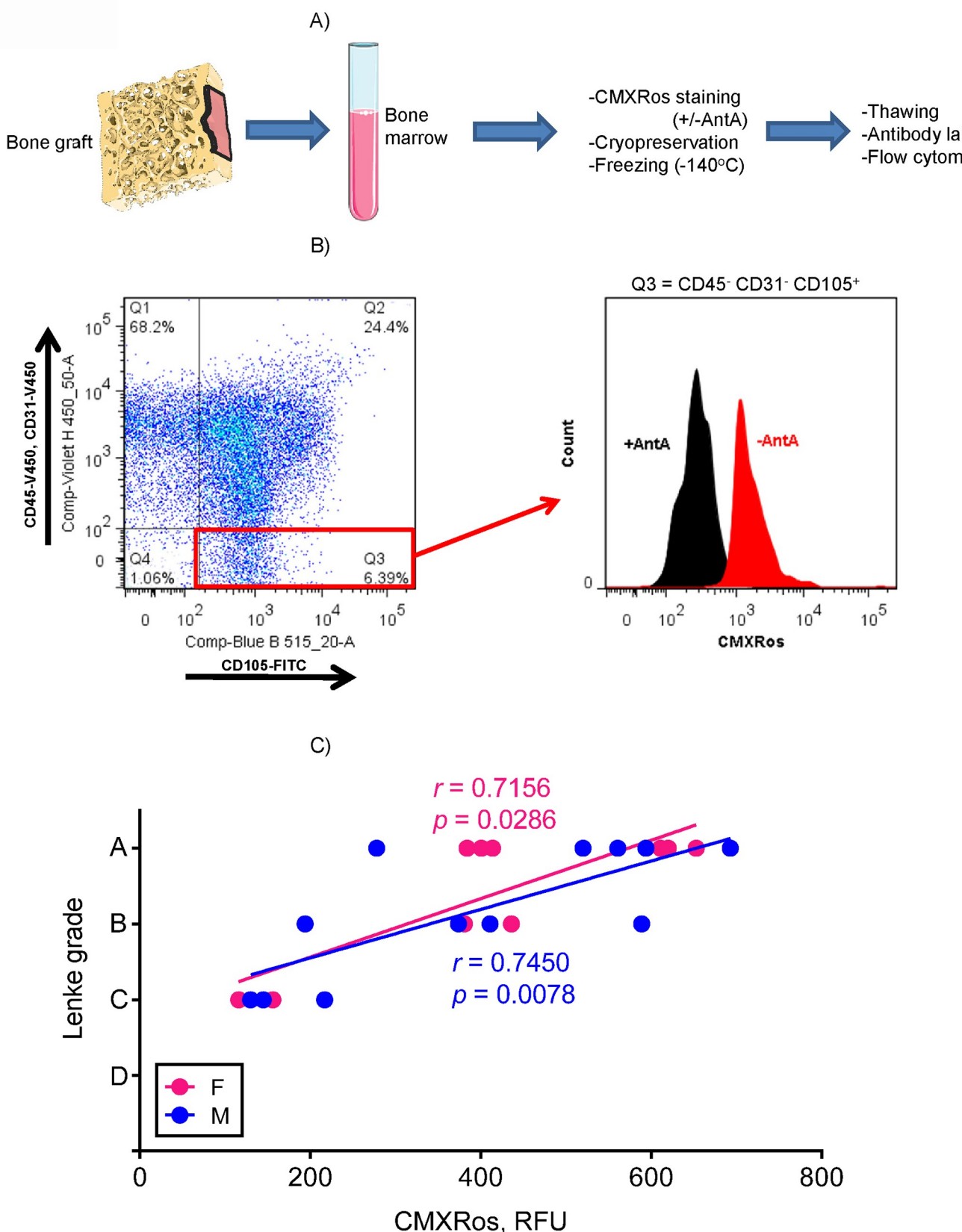

**Fig 3. BMSC mitochondrial oxidative metabolism correlates with better spinal fusion outcomes.** A) Mitochondrial OxPhos function in bone marrow cells was measured with the potentiometric fluorescent probe, CMXRos; B) BMSCs were identified as CD45- CD31- CD105+ population. Antimycin A (AntA) was used as a negative control. Difference between CMXRos signal without AntA and with AntA was calculated as a measure of mitochondrial membrane potential; C) CMXRos signal shows strong correlation (Spearman test) with the spinal fusion radiographic Lenke score.

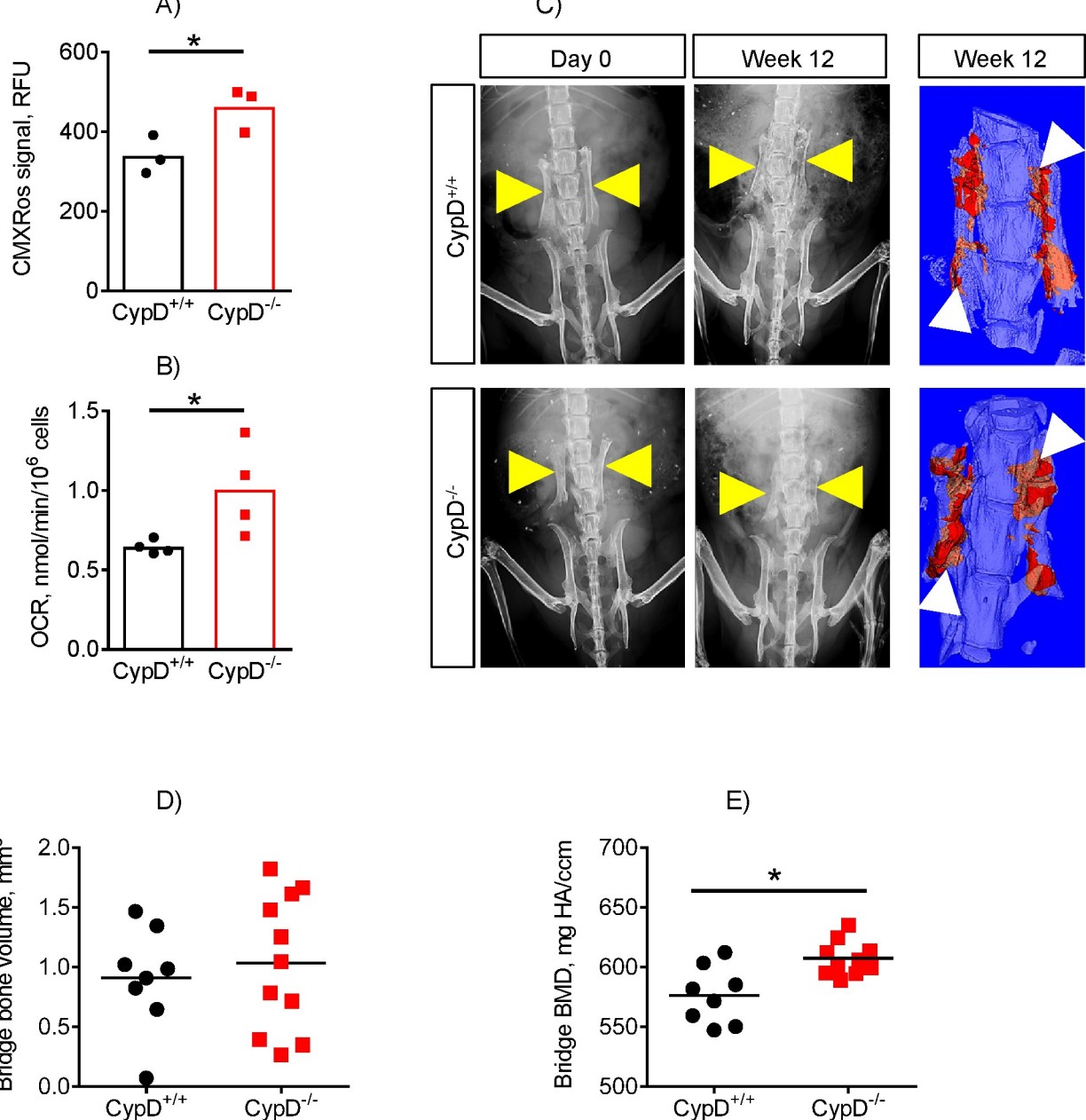

**Fig 4. Mice with higher oxidative metabolism show better mineralization of spinal fusion grafts.** A) When compared to their control (CypD$^{+/+}$) littermates, BMSCs from CypD$^{-/-}$ mice have higher oxidative metabolism measured with both CMXRos (A) and via oxygen consumption rate (OCR) assay using Seahorse XF technology (B); C) X-ray and micro-CT images of spinal grafts at D0 and at 12 wks post-surgery. Yellow arrowheads indicate grafts on X-ray and white arrowheads indicate newly formed bone (colored red) in micro-CT scans. Newly formed bone volume (D) and bone mineral density (BMD, E) were derived from micro-CT analysis of new bone formation at the fusion site. Plots show actual data points and calculated Means. *, $p < 0.05$ (*t*-test).

## Discussion

This study shows that patients with high BMSC oxidative metabolism had more favorable outcomes in spinal fusion surgery than those patients with low BMSC oxidative metabolism. This correlative finding was mechanistically confirmed using a genetic mouse model of increased oxidative metabolism undergoing spinal fusion. We have also shown that the colony forming ability and osteogenicity of BMSCs *in vitro* do not correlate with spinal fusion outcomes assessed by the Lenke grade. Spinal fusion outcomes also did not correlate with age in the studied cohort.

There is still much work to be done regarding the role of oxidative metabolism in osteolineage cells and bone healing, and how this knowledge could be clinically relevant. The Lenke grading scale for fusions is highly subjective and may not be consistent between multiple surgeons or institutions. Additionally, this work is without a direct intervention to demonstrate specifically how mitochondrial function in BMSCs could lead to better bone healing. The specific mechanisms of how mitochondrial function promotes osteogenicity have yet to be fully elucidated and is the target of our current and future work. With regards to this, we have published a report showing that activation of mitochondrial OxPhos in osteoprogenitors leads to acetylation and activation of β-catenin by providing acetyl-coenzyme A, and thus promoting osteogenesis [63]. We currently lack clear radiographic and biomechanical testing criteria for spinal fusion outcomes in a mouse model. Using micro-CT, we were able to detect new bridging bone, the amount of this new bone, and its BMD. We observed higher BMD of the fusion bridge in the mice that received grafts containing BMSCs with more active mitochondria. How the increased bridge BMD translates into biomechanical stability of fused spine in mice is still unclear. However, it is reasonable to assume that higher BMD indicates stronger bridging bone and more stable fusion. Finally, in our mouse studies, we used male mice only to exclude possible effects of estrogen on BMSC oxidative metabolism [64–66].

Of note, there are a few medications as well as behavioral factors (e.g. smoking) listed in Table 1 that could have potential effects on bone quality and spinal fusion outcome (Table 3). In particular, nicotine use has been reported to increase the rate of perioperative complications, including pseudoarthrosis [67]. However, the effect of nicotine is complex, perhaps dose-dependent, and may not always be detrimental [68]. Non-steroidal anti-inflammatory drug (NSAID) use may be associated with delayed bone healing [69]. Long term use of the anti-epileptic carbamazepine has been associated with bone disease, evidenced by

**Table 3. Potential effects of medications on bone.**

| Medications | Potential effect on bone |
| --- | --- |
| Carbamazepine | Possible adverse effect |
| Cetirizine | No effect |
| Cimetidine | No effect |
| Clonazepam | No effect |
| Fenofibrate | No effect |
| Furosemide | No effect |
| Hydrochlorothiazide | Possible beneficial effect |
| Hydralazine | No effect |
| Ibuprofen | Possible adverse effect |
| Levothyroxine | No Effect |
| Methocarbamol | No Effect |
| Metoprolol | No Effect |
| Nicotine | Possible adverse effect |

biomechanical abnormalities and decreased bone mineral density [70]. Hydrochlorothiazide use has been shown to produce small positive benefits on cortical bone density [71]. Future research is needed to further elucidate the effects of these and other drugs on bone quality and spinal fusion outcome.

To our knowledge, this is the first study to show a correlation between mitochondrial oxidative function in BMSCs and spinal fusion outcome in humans, and to provide initial evidence of causation using a genetic mouse model. This information could potentially be used to prospectively assess the need for additional clinical treatments, such as BMP-2, for spinal fusion patients. Oxidative metabolism in BMSCs varies among individuals due to genetic and environmental factors. Epigenetic modifications, including histone acetylation, have been shown to affect cellular metabolism [72,73]. In fact, obese patients with ventricular cardiac dysfunction displayed mitochondrial hyperacetylation [74]. Our study suggests that such variations may underline the observed differences in outcomes of spinal fusion between patients. Additionally, this study opens new research questions regarding whether improving mitochondrial function of BMSCs prior to grafting would improve surgical outcomes.

## Acknowledgments

We would like to thank Mr. Michael Thullen and Dr. Hani Awad from the Center for Musculoskeletal Research for their help with micro-CT, Dr. Charles O Smith for his help with figures, clinical coordinators from the University of Rochester Department of Orthopaedics, and Dr. Pamela Robey from NIH for valuable advice.

## Author Contributions

**Conceptualization:** Roman A. Eliseev.

**Data curation:** Laura C. Shum, Alex M. Hollenberg, Avionna L. Baldwin, Brianna H. Kalicharan, Paul T. Rubery, Addisu Mesfin.

**Formal analysis:** Laura C. Shum, Alex M. Hollenberg, Avionna L. Baldwin, Noorullah Maqsoodi, Paul T. Rubery, Addisu Mesfin, Roman A. Eliseev.

**Funding acquisition:** Laura C. Shum, Addisu Mesfin, Roman A. Eliseev.

**Investigation:** Laura C. Shum, Alex M. Hollenberg, Avionna L. Baldwin, Brianna H. Kalicharan, Noorullah Maqsoodi, Addisu Mesfin.

**Methodology:** Laura C. Shum, Alex M. Hollenberg, Brianna H. Kalicharan, Paul T. Rubery, Addisu Mesfin.

**Resources:** Roman A. Eliseev.

**Supervision:** Roman A. Eliseev.

**Writing – original draft:** Laura C. Shum, Alex M. Hollenberg, Roman A. Eliseev.

**Writing – review & editing:** Laura C. Shum, Alex M. Hollenberg, Brianna H. Kalicharan, Noorullah Maqsoodi, Addisu Mesfin, Roman A. Eliseev.

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
