## [Decision Letter · Decision Letter 0]

1 Sep 2020

PONE-D-20-23934

Role of oxidative metabolism in osseointegration during spinal fusion

PLOS ONE

Dear Dr. Eliseev,

Thank you for submitting your manuscript to PLOS ONE. After careful consideration, we feel that it has merit but does not fully meet PLOS ONE’s publication criteria as it currently stands. Therefore, we invite you to submit a revised version of the manuscript that addresses the points raised by an expert reviewer and editorial remarks as noted below during the review process.

We look forward to receiving your revised manuscript.

Kind regards,

Dr. Sakamuri V. Reddy

Academic Editor

PLOS ONE

Journal Requirements:

3. We note you have included a table to which you do not refer in the text of your manuscript. Please ensure that you refer to Table 2 in your text; if accepted, production will need this reference to link the reader to the Table.

Additional Editor Comments (if provided):

The authors have identified the oxidative function due to mitochondrial membrane potential correlation with spinal fusion. Also, the mouse models of spinal fusion and cyclophilin D knock-out (CypD KO) with an improved mitochondrial function had higher mineralization of the spine fusion. These data suggested a role for oxidative metabolism in bone marrow stromal cells (BMSC) in spinal fusion process. Specific comments include, the Introduction section is written briefly. They may note the related info ex., mechanisms underlying the biological enhancement of spinal fusion in spinal fractures/degenerative diseases. Methods-please include citations for methods available, ex., CFU assay, CypD-/- mice, Seahorse metabolic analysis etc. Results (page 12; line 219) It is unclear how they note BMSCs to form colonies (CFU assay) when it is not a method to assess BMSC quality and other hematopoietic lineage cells for CFU-GM colonies. The authors should clarify the rationale for these experiments and discussed appropriately. Discussion section is written briefly with one citation. They should include other factors like BMP they have noted, processes influencing oxidative potential of mitochondria and spine fusion/repair process.

Reviewers' comments:

Reviewer's Responses to Questions

**Comments to the Author**

1. Is the manuscript technically sound, and do the data support the conclusions?

Reviewer #1: Partly

2. Has the statistical analysis been performed appropriately and rigorously? 

Reviewer #1: I Don't Know

3. Have the authors made all data underlying the findings in their manuscript fully available?

Reviewer #1: No

4. Is the manuscript presented in an intelligible fashion and written in standard English?

Reviewer #1: Yes

5. Review Comments to the Author

Reviewer #1: This study was about to determine the impact of BMSC oxidative metabolism on osseointegration of the graft during spinal fusion. Following are the several key points that are required to be considered by the authors for thorough revisions.

1.The authors should include more information that clarifies and justifies their choice of methods.

2. It is important to prove that the improved mitochondrial function has direct relationship with improved spinal fusion outcomes since it could also have vital influences on the overall metabolism of individuals.

3. Do the different medications for other diseases treatment mentioned in table 1 have notable influences on the outcomes? Please give that part in the discussion.

4. Because oxidative metabolism is the main topic in the literature, please give more illustrations about “Oxidative metabolism varies among individuals due to genetic and environmental factors” in the discussion part. Is any obvious environmental factor related to the subjects?

5. The article misses important references in several presentations, such as line67, line253.

6. Please give interpretations about using cells treated with antimycin A as a negative control and about the way to differentiate cells in the discussion part.

6. PLOS authors have the option to publish the peer review history of their article (what does this mean?). If published, this will include your full peer review and any attached files.

Reviewer #1: No

---

## [Author Response · Author response to Decision Letter 0]

22 Oct 2020

October 21, 2020

PONE-D-20-23934

Role of oxidative metabolism in osseointegration during spinal fusion

PLOS ONE

Dear Dr. Reddy,

We would like to thank the Editors and the Reviewers for their excellent comments. We have done our best to address these comments and revise our manuscript. All the changes in the text of our revised manuscript are tracked. The clean (no markup) final version is also included. Below are our point-by-point responses: 

Editorial Comments:

2. “Please provide additional details regarding participant consent.”

We have added the required information in the Methods section and in the Ethics Statement online.

3. “Please ensure that you refer to Table 2 in your text”

Table 2 is now referred to in the text of the Methods under ‘Lenke Grading’

Additional Editor Comments:

“The Introduction section is written briefly. They may note the related info ex., mechanisms underlying the biological enhancement of spinal fusion in spinal fractures/degenerative diseases.”

We expanded our Introduction to include the suggested information. 

“Methods-please include citations for methods available…”

Citations were added to the Methods.

“Results (page 12; line 219) It is unclear how they note BMSCs to form colonies (CFU assay) when it is not a method to assess BMSC quality…”

This issue was addressed and text modified where indicated.

“Discussion section is written briefly with one citation. They should include other factors.”

The Discussion was considerably extended with inclusion of suggested information and citations.

Reviewer # 1:

1. “The authors should include more information that clarifies and justifies their choice of methods.”

We modified our Methods and Results sections to include this justification

2. “It is important to prove that the improved mitochondrial function has direct relationship with improved spinal fusion outcomes since it could also have vital influences on the overall metabolism of individuals.”

This is an excellent point. Mitochondrial function undoubtedly influences overall metabolism. Our patient data are most likely reflective of not only BMSC but also overall metabolism and systemic properties. Please note that we studied the cases where autografts were used so both BMSCs and the fusion bed had likely the same properties. This is why to model this situation in mice, we used both host and donor of the same phenotype, e.g CypD-/- graft in CypD-/- donor. We modified our Discussion section to address this.

This also provides a rationale for a separate study focusing on donor BMSC metabolism if it is different from the recipient’s metabolism and whether the donor cells can be manipulated to improve fusion.

3. “Do the different medications for other diseases treatment mentioned in table 1 have notable influences on the outcomes?”

We included a new Table 3 to address this comment and modified our Discussion section accordingly. We also added several references on possible effects of these treatments on bone.

4. “Because oxidative metabolism is the main topic in the literature, please give more illustrations about “Oxidative metabolism varies among individuals due to genetic and environmental factors” in the discussion part.”

The Discussion was modified to include more information about the effect of genetic and environmental factors.

5. “The article misses important references in several presentations”

New references were added where indicated

6. “Please give interpretations about using cells treated with antimycin A as a negative control and about the way to differentiate cells in the discussion part.”

The information on the use of antimycin A was added to both Methods and Results. With regards to cell differentiation, we would like to note that cells were not specifically induced to differentiate. We used CFU-O (alkaline phosphatase positivity) assay in undifferentiated CFU that start to spontaneously express alkaline phosphatase after overgrowing within the colony. This method has been frequently used in various studies although CFU-Os in osteogenic media is also frequently used. 

We hope that with these corrections, our manuscripts is now acceptable for publication in PLoS One journal.

Sincerely,

Roman Eliseev, MD, PhD

---

## [Decision Letter · Decision Letter 1]

26 Oct 2020

Role of oxidative metabolism in osseointegration during spinal fusion

PONE-D-20-23934R1

Dear Dr. Eliseev,

We’re pleased to inform you that your manuscript has been judged scientifically suitable for publication and will be formally accepted for publication once it meets all outstanding technical requirements.

Kind regards,

Dr. Sakamuri V. Reddy

Academic Editor

PLOS ONE

---

## [Editor Report · Acceptance letter]

28 Oct 2020

PONE-D-20-23934R1 

Role of oxidative metabolism in osseointegration during spinal fusion 

Dear Dr. Eliseev:

I'm pleased to inform you that your manuscript has been deemed suitable for publication in PLOS ONE. Congratulations! Your manuscript is now with our production department. 

Kind regards, 

on behalf of

Dr. Sakamuri V. Reddy 

Academic Editor

PLOS ONE